# Catalytic Hydrodeoxygenation of Guaiacol to Cyclohexanol over Bimetallic NiMo-MOF-Derived Catalysts

**Minghao Zhou** [1,*]**, Fei Ge** [1]**, Jing Li** [2]**, Haihong Xia** [2]**, Junli Liu** [2]**, Jianchun Jiang** [2]**, Changzhou Chen** [2,3]**, Jun Zhao** [3,*] **and Xiaohui Yang** [2,*]

1   School of Chemistry and Chemical Engineering, Yangzhou University, Yangzhou 225002, China; ydxff41@163.com
2   Key Laboratory of Biomass Energy and Material, Institute of Chemical Industry of Forest Products, Chinese Academy of Forestry, Nanjing 210042, China; lijing0803@126.com (J.L.); xiahaihong87@126.com (H.X.); xya1436432986@163.com (J.L.); tangcj0407@163.com (J.J.); changzhou_chen@163.com (C.C.)
3   Department of Biology, Institute of Bioresource and Agriculture, Hong Kong Baptist University, Kowloon Tong, Hong Kong 999077, China
*   Correspondence: zhouminghao@yzu.edu.cn (M.Z.); zhaojun@hkbu.edu.hk (J.Z.); yxh@icifp.cn (X.Y.)

**Abstract:** Lignin is an attractive renewable source of aromatics with a low effective hydrogen to carbon ratio ($H/C_{eff}$). The catalytic hydrodeoxygenation (HDO) of lignin-derived model compounds is a key strategy for lignin upgrading. In this work, the HDO of guaiacol, a typical lignin-derived compound, was carried out over metal–organic framework (MOF)-derived Ni-based catalysts. A monometallic Ni-MOF catalyst and different ratios of bimetallic NiMo-MOF catalysts were synthesized by a hydrothermal process, followed by a carbonization process. Among these catalysts, Ni3Mo1@C exhibited an excellent catalytic performance, affording a guaiacol conversion of 98.8% and a cyclohexanol selectivity of 66.8% at 240 °C and 2 MPa $H_2$ for 4 h. The addition of Mo decreased the particle size of the spherical structure and improved the dispersion of metal particles. The synergistic effect between Ni and Mo was confirmed by various means, including ICP, XRD, SEM, TEM, and $NH_3$-TPD analyses. In addition, the effect of the reaction temperature, time, and $H_2$ pressure during the HDO process is discussed in detail.

**Keywords:** guaiacol; hydrodeoxygenation; Ni-based bimetallic catalysts; MOF

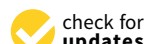



## 1. Introduction

In view of the sharp depletion of fossil fuels and increasing energy demands, tremendous efforts have been contributed to finding alternative sources. As the most abundant renewable alternative to fossil feedstock, the conversion of biomass to liquid fuels or fine chemicals has been widely investigated in recent years [1–6]. Although cellulose and hemicellulose have been well studied in the field of conversion to valued-added chemicals and biofuels, lignin, the second largest component in lignocellulosic biomass, is treated as a waste of biorefining and is burnt to generate heat [7–10]. Thus, the complete utilization of lignin is still a challenge. In fact, lignin is a three-dimensional amorphous polymer and consists of highly cross-linked phenylpropane units (p-Coumaryl, coniferyl, and sinapyl alcohol) connected by C-O and C-C bonds, which account for 15–30 wt% of its mass and approximately 40% of the energy of lignocellulose, offering an opportunity to convert it into value-add chemicals and liquid fuel [11–14]. Through a fast pyrolysis process, lignin can be converted into bio-oil. However, due to the high oxygen content and low $H/C_{eff}$ (~0.6) of crude bio-oil, it has a lower heating value and poorer thermal stability compared with transport fuels, which makes it impossible to be used on a large scale [15–17]. Therefore, to improve the quality of bio-oil, a deoxygenation process must be implemented to boost the H:C and C:O ratios for upgrading bio-oil to a transport fuel.

In recent decades, it has been reported that catalytic hydrodeoxygenation (HDO) is a promising method to eliminate oxygen [18,19]. The catalytic hydrodeoxygenation of lignin-derived model compounds and lignin exhibits good conversion and a high selectivity of products, such as cyclohexane and cyclohexanol, with an H/Ceff of 2.0 and 1.667 [20,21]. According to previous studies, heterogeneous catalysts, consisting of precious metals and non-precious metals, and homogenous catalysts have been studied. Precious metal catalysts including Pt [22], Pd [23], Re [24], and Rh [25] have high activity and stability for the HDO of lignin and lignin-derived model compounds, while the expensive price limits their large-scale application [26]. Moreover, due to the high cost and difficulty of catalyst and product separation, homogenous catalysts are not a good choice [27]. Therefore, non-precious mental catalysts, such as Ni [28], Co [29], Mo [30], and Cu [31], attract researchers' attention. Ni-based catalysts, in particular, have received more attention because of their relatively high catalytic activity and low price. He et al. [32] reported the HDO of aryl ethers over an Ni/SiO$_2$ catalyst at 120 °C with 6 bar H$_2$, obtaining aromatic and aliphatic molecules as well as cyclohexanol. Kordouli et al. [33] investigated the HDO of phenol over reduced Ni/ASA and NiMo/ASA in a flow reactor at 310 °C under 4 MPa extra hydrogen and revealed the promoting action of Mo in NiMo/ASA.

As an excellent newcomer in the field of catalysis, metal–organic frameworks (MOFs) have attracted increasing attention due to their large surface area, clear porous structures, low density, and tunable active site, which promotes catalytic activity. In particular, the combination of MOFs with Ni-based catalysts shows an excellent hydrodeoxygenation ability [34–36]. For instance, Liu et al. [36] synthesized an array of Ni/C-derived catalysts using different MOFs as the metal precursors, which showed good activity with 98.88% conversion and a 4.61% cracking ratio for the hydrotreatment of triolein into green diesel. Chen et al. [37] prepared metal–organic framework-derived Ni-La@C catalysts, exhibiting perfect catalytic activity for the hydrogenolysis of lignin dimmers. Han et al. [38] utilized MOFs as precursors to immobilize ultrafine Ni and Co on SiO$_2$ supports, which showed good activity and stability for the liquid-phase hydrogenation of benzene at a lower temperature. Yang et al. [39] fabricated Ni/C and bimetallic NiCo@C catalysts starting from microporous MOFs for an efficient conversion of poplar lignin to monophenols. The bimetallic catalyst Ni$_x$Co$_{1-x}$@C, forming an alloy structure, improved the catalytic activity of lignin depolymerization due to the synergistic effect. Different biomass-derived compounds have been catalyzed by Ni-based, MOF-derived catalysts. Therefore, further research in this field should be conducted [40,41].

Owing to phenolic compounds accounting for a large amount of bio-oil, in this work, guaiacol was firstly utilized as a lignin-derived phenolic model compound, as it contained the representative functional groups such as hydroxyl (-OH) and methoxy (-OCH$_3$) [42,43]. Ni-based metal-organic frameworks (Ni-MOF and NiMo-MOF) were successfully prepared. Metal-organic framework-derived Ni/C and bimetallic NiMo@C catalysts with a high loading amount and good dispersity were obtained from MOF precursors. The catalytic hydrodeoxygenation of guaiacol was carried out in isopropanol under a hydrogen atmosphere over Ni@C and NiMo@C. The promoting action of Mo was observed due to the higher cyclohexanol yield. The synergistic capability of Ni and Mo was discussed through X-ray diffraction (XRD), scanning electronic microscopy (SEM), transmission electron microscopy (TEM), inductively coupled plasma emission spectrometry (ICP), NH$_3$-temperature-programmed desorption (NH$_3$-TPD), and pyridine-IR (Py-IR) analyses. In addition, different temperatures, reaction times, and hydrogen pressures were investigated to explore the optimal reaction conditions. This work may offer an efficient strategy for the hydrodeoxygenation of lignin and upgrading of bio-oil.

## 2. Result and Discussion

### 2.1. Catalyst Characterization

As the ICP analysis presented in Table 1, the Ni@C catalyst showed a high Ni content of 59.88 wt%. For the bimetallic NiMo@C catalysts, the total metal Ni and Mo content was still

high: 49.12 wt% of Ni and 10.08 wt% of Mo for Ni5Mo1@C; 48.89 wt% and 12.17 wt% of Mo for Ni4Mo1@C; 44.86 wt% of Ni and 15.48 wt% of Mo for Ni3Mo1@C; and 40.91 wt% of Ni and 21.98 wt% of Mo for Ni2Mo1@C. The real Ni/Mo mole ratio in catalysts was 4.87, 4.02, 2.89, and 1.86, respectively, which was very close to the theoretical values. These results indicated that Ni@C and different ratios of NiMo@C catalysts were successfully prepared.

**Table 1.** ICP analysis of Ni@C and NiMo@C with different Ni/Mo ratio.

|                    | Ni@C  | Ni5Mo1@C | Ni4Mo1@C | Ni3Mo1@C | Ni2Mo1@C |
|--------------------|-------|----------|----------|----------|----------|
| Ni amount (wt.%)   | 59.88 | 49.12    | 48.89    | 44.86    | 40.91    |
| Mo amount (wt.%)   | /     | 10.08    | 12.17    | 15.48    | 21.98    |
| Ni/Mo mole ratio   | /     | 4.87     | 4.02     | 2.89     | 1.86     |

The phase structure of Ni@C and NiMo@C with different Ni/Mo ratios was studied through XRD analysis, as shown in Figure 1. The XRD pattern of the Ni@C catalyst showed four diffraction peaks at 39.5°, 42.3°, 45.0°, and 59. 5°, which can be assigned to the (110), (006), (113), and (116) planes of $Ni_3C$ [1]. There existed two obvious diffraction peaks at $2\theta = 44.3°$ and 51.7° in Ni2Mo1@C, Ni3Mo1@C, Ni4Mo1@C, and Ni5Mo1@C, which can be ascribed to the (111) and (200) planes of metal Ni. The diffraction peaks of $Ni_3C$ were undetected in Ni3Mo1@C and Ni2Mo1@C, and thus suggested that the addition of Mo could promote the formation of metal Ni and prevent the appearance of $Ni_3C$. Above all, the results indicated that the introduction of Mo strengthened the interaction between metal Ni and Mo in NiMo@C catalysts, and the formation of NiMo alloy may be be confirmed due to the changes of the nickel's crystal structure.

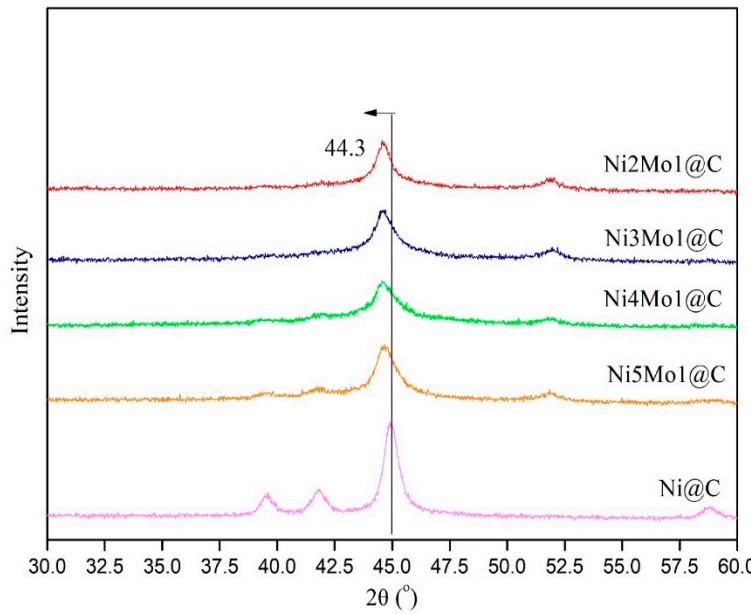

**Figure 1.** XRD patterns of Ni@C, Ni2Mo1@C, Ni3Mo1@C, Ni4Mo1@C, and Ni5Mo1@C.

To obtain the micro-structure features of different catalysts, Ni@C and NiMo@C with different Ni/Mo ratios were investigated by SEM. As presented in Figure 2, the spherical structure was observed in Ni@C and in the different ratios of NiMo@C catalysts. With the increasing amount addition of Mo in NiMo@C, the particle size of the spherical structure increased gradually, which was in alignment with the TEM results. In addition, metal Ni and Mo were still evenly distributed, which was observed by element mappings. The element mappings also clearly exhibited that the content of Ni increased and the content of Mo decreased in catalysts with Ni/Mo ratios from 2:1 to 5:1.

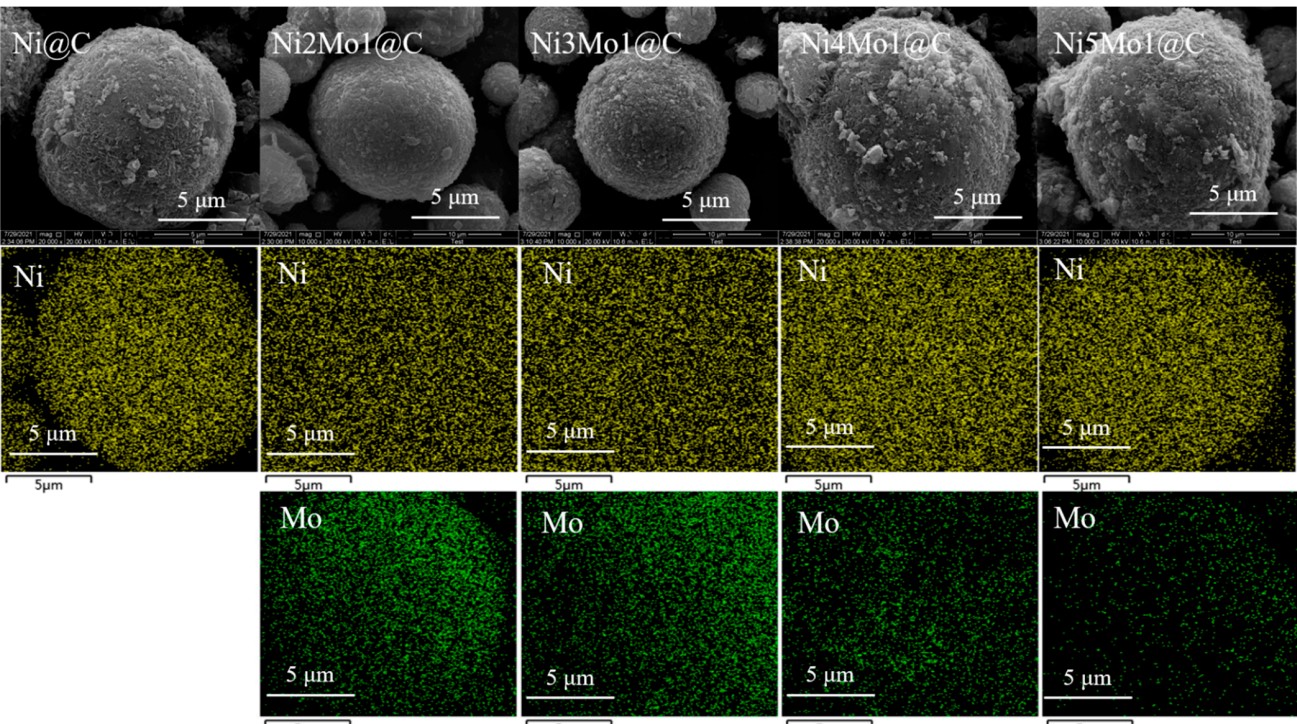

**Figure 2.** SEM images and elemental mappings of Ni@C, Ni2Mo1@C, Ni3Mo1@C, Ni4Mo1@C, and Ni5Mo1@C.

TEM analysis of Ni@C and different Ni/Mo ratios of NiMo@C was also conducted to investigate the effect of the addition of Mo on particle dispersion and particle size. The TEM images and particle size distribution of different catalysts are presented in Figure 3. The TEM results also showed that metal Ni and Mo were uniformly dispersed in catalysts, and the particle size of Ni and Mo metals were loaded, which corresponded with the SEM-mapping results. It was observed that the main particle size and particle distribution varied with the introduction of Mo to the NiMo@C catalysts. The average particle size of Ni in monometallic Ni and bimetallic NiMo catalysts ranged as follows: Ni@C (24.27 nm) > Ni2Mo1@C (23.39 nm) > Ni5Mo1@C (7.18 nm) > Ni3Mo1@C (6.51 nm) > Ni4Mo1@C (6.50 nm). The smaller average size was found in bimetallic NiMo@C catalysts due to the promoting effect of Mo, which strengthened the interaction between metal Ni and Mo to reduce the particle size. Surprisingly, the average particle size of Ni2Mo1@C was 23.39 nm, which was bigger than Ni3Mo1@C, Ni4Mo1@C, and Ni5Mo1@C. This was attributed to a higher amount of added Mo, which resulted in metal aggregation. The metal aggregation phenomenon was clearly seen in the TEM micrograph of Ni2Mo@C. As we all know, a smaller particle size and a good dispersion can promote a hydrodeoxygenation reaction, which was described as the particle size effect [44]. Thus, the introduction of Mo decreased the particle size, and then promoted the HDO reaction of guaiacol to cyclohexanol.

The acidity of Ni@C and different ratios of the NiMo@C catalysts were studied by $NH_3$-TPD and Py-IR. $NH_3$-TPD was carried out to investigate the total acidity in the Ni@C and bimetallic NiMo@C catalysts. As seen in the results listed in Table 2, the total acidity for NiMo@C increased with the addition of Mo in the catalysts, and the total acidity for Ni5Mo1@C, Ni4Mo1@C, and Ni3Mo1@C was improved to 1.44 mmol/g $NH_3$, 1.79 mmol/g $NH_3$, and 2.01 mmol/g $NH_3$, respectively, which was comparable with Ni@C (0.55 mmol/g $NH_3$). The total acidity was up to its highest at 2.01 mmol/g $NH_3$ in Ni3Mo1@C, and then decreased to 1.88 mmol/g $NH_3$ when the amount of Mo increased to Ni2Mo1@C. Thus, Ni3Mo1@C could provide the most acid sites during the following hydrogenation of guaiacol to promote guaiacol conversion to cyclohexanol. Py-IR was also conducted to study Brønsted acid and Lewis acid distribution in different catalysts. The trends of

Brønsted acid and Lewis acid were in accordance with total acidity analysis results. For Ni2Mo1@C, Ni3Mo1@C, Ni4Mo1@C, and Ni5Mo1@C, the Brønsted acid was improved to 225.36 μmol/g, 275.06 μmol/g, 248.81 μmol/g, and 190.02 μmol/g, respectively, compared with Ni@C (142.32 μmol/g); and the Lewis acid was 6.00 μmol/g, 6.88 μmol/g, 5.68 μmol/g, and 4.56 μmol/g, respectively, compared with Ni@C (3.25 μmol/g). The Ni3Mo@C catalyst had the highest level of Brønsted acid and Lewis acid, which could offer the most acid sites during the HDO of guaiacol. Although metal Mo could enhance the acidity of the NiMo@C catalyst, an excessive addition of Mo could not continuously increase the acidity. In summary, the Ni3Mo1@C catalyst exhibited the strongest acidity among these catalysts, which indicated that the synergistic effect between metal Ni and Mo could result in a strong electron transfer to improve the acidity and increase the number of acid sites.

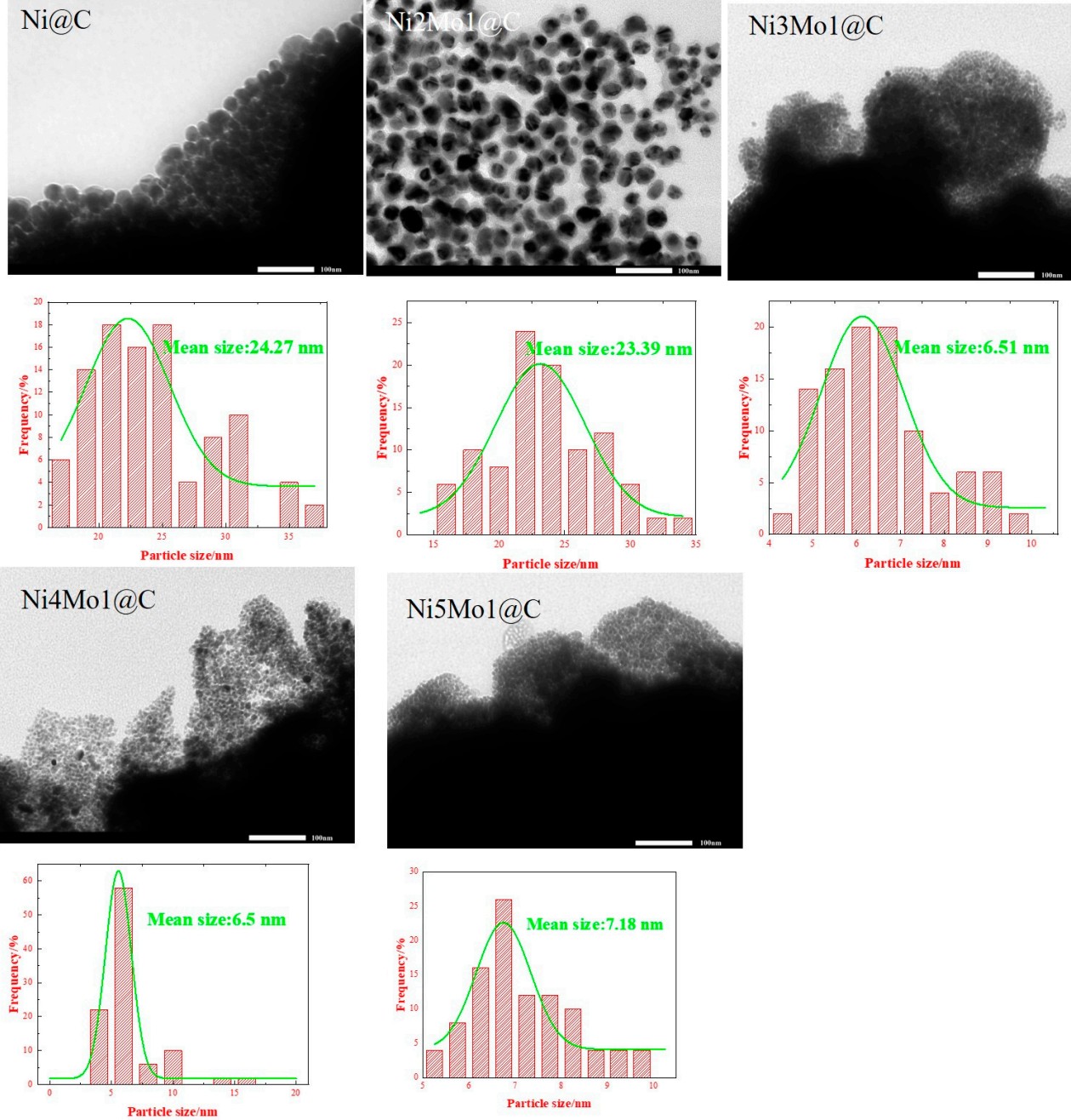

**Figure 3.** The TEM micrographs and particle size distribution of Ni@C, Ni2Mo1@C, Ni3Mo1@C, Ni4Mo1@C, and Ni5Mo1@C.

**Table 2.** Total acidity and Brønsted acid and Lewis acid distribution in Ni-based catalysts derived from Ni-MOF and NiMo-MOF.

|  | Ni@C | Ni5Mo1@C | Ni4Mo1@C | Ni3Mo1@C | Ni2Mo1@C |
|---|---|---|---|---|---|
| Total acidity [a] (mmol/g NH$_3$) | 0.55 | 1.44 | 1.79 | 2.01 | 1.88 |
| Lewis acid [b] (μmol/g) | 142.32 | 190.02 | 245.81 | 275.06 | 225.36 |
| Bronsted acid [b] (μmol/g) | 3.25 | 4.56 | 5.68 | 6.88 | 6.00 |

[a] detected by NH$_3$-TPD; [b] detected by pyridine-IR.

The valence states of metal Ni and Mo were investigated by XPS, as shown in Figure S1. The peaks located at 869.7 and 852.4 eV corresponded to the characteristic peaks of Ni 2p1/2 and Ni 2p3/2, which were assigned to metallic Ni. The peaks located at 856.1 eV corresponded to the characteristic peaks of Ni 2p3/2, and they were assigned to nickel oxide. A small amount of nickel oxide appeared in the Ni2Mo1@C catalyst, indicating that the addition of excessive Mo leads to the oxidation of metal Ni.

*2.2. Catalytic Hydrodeoxygenation of Guaiacol*
2.2.1. Effect of Metal Composition in Catalysts

Herein, the catalytic hydrodeoxygenation of guaiacol was carried out over the monometallic Ni@C catalyst and bimetallic NiMo@C catalysts with different Ni/Mo ratios for the initial screening of the catalysts. The guaiacol conversion and product selectivity are presented in Figure 4. As shown in Figure 4, all the conversions of guaiacol were almost 100% for the MOF-derived, Ni-based catalysts. When the monometallic Ni@C catalyst was used in the HDO process, the conversion of guaiacol was enhanced to 97.8%, while in a previous study, Chen et al. carried out the HDO of guaiacol over Ni/CNT with a very low guaiacol conversion (around 40%) [45], and there existed a slight increase of guaiacol conversion when the HDO reaction was conducted over bimetallic NiMo@C catalysts. These results indicated that metal–organic frameworks (MOFs) can promote high catalytic activity due to their high density of uniform and tunable active and MOF-derived, Ni-based catalysts, which exhibit a good hydrodeoxygenation capacity. The catalyst of Ni@C, phenol, coming from the demethoxylation of guaiacol, was one of the main products; however, phenol disappeared, owing to the further hydrogenation to cyclohexanol when the HDO reactions were carried out over bimetallic NiMo@C catalysts. Therefore, the promoting action of Mo maybe be confirmed. It was noteworthy that when Ni2Mo1@C was applied in the HDO process, phenol was still the main product, with a high selectivity of 80%, while phenol was undetected and cyclohexanol and cyclohexane were the main products when the Ni/Mo ratios were 5:1, 4:1, and 3:1, respectively. The H/C$_{eff}$ of phenol (~0.67) is lower than cyclohexanol, with an H/C of 1.67, indicating that the hydrodeoxygenation capacity of Ni2Mo1@C is lower than Ni5Mo1@C, Ni4Mo1@C, and Ni3Mo1@C. The result was attracted to a bigger particle size and metal particles aggregation resulting from a large amount of Mo added in the Ni2Mo1@C catalyst, which was confirmed by TEM micrographs. The NiMo@C catalysts with the ratios of 5:1, 4:1, and 3:1 showed a similar selectivity of cyclohexanol (around 67%). As is known, a smaller size of particle diameter and a better dispersion of metal particles are beneficial to catalytic hydrodeoxygenation. Therefore, the introduction of a suitable amount of Mo can decrease the particle size of NiMo@C catalysts, and the interaction between Ni and Mo can improve the metal dispersion to lead to better catalytic activity in this HDO process, which is regarded as a synergistic effect. Moreover, the particle diameter of Ni5Mo1@C, Ni4Mo1@C, and Ni3Mo1@C is smaller compared with Ni2Mo1@C. Thus, owing to particle size effect, Ni2Mo1@C had lower hydrogenation activity. In addition to particle size, acidity also played an important role in the guaiacol HDO process. Ni3Mo1@C exhibited a high acidity amount, which showed a high selectivity of cyclohexanol, with a low selectivity of other byproducts. Taken together,

these factors could affect the catalytic capacity during the guaiacol HDO process, and as is shown in Figure 4, promoting the action of Mo can greatly enhance the selectivity of cyclohexanol. The best performance was found over Ni3Mo1@C; therefore, further studies were conducted to seek the optimal conditions.

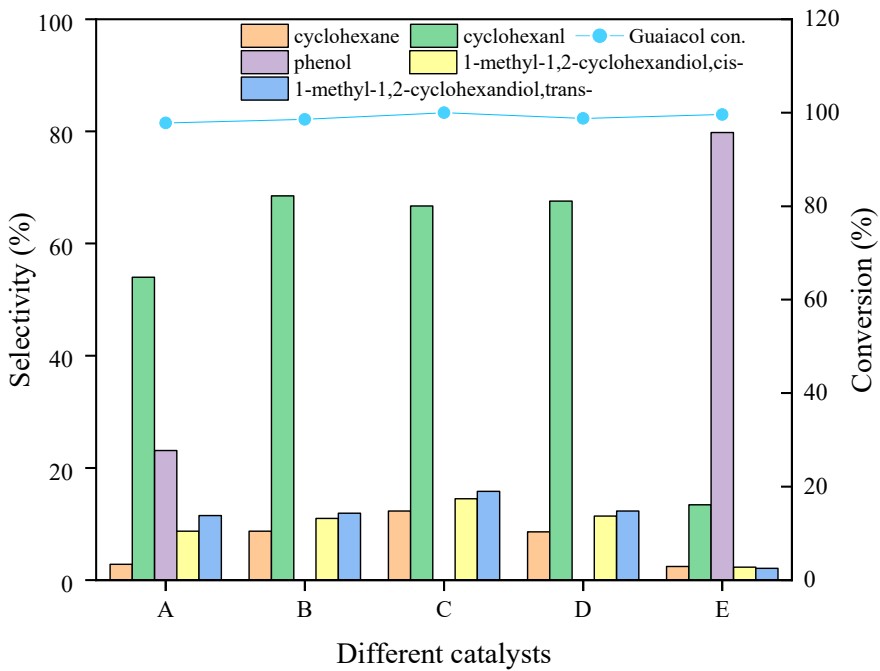

**Figure 4.** Guaiacol conversion and product distribution over different catalysts. Reaction conditions: T = 240 °C, p = 2 MPa, and t = 4 h; catalysts: (A) Ni@C, (B) Ni5Mo1@C, (C) Ni4Mo1@C, (D) Ni3Mo1@C, and (E) Ni2Mo1@C.

### 2.2.2. Product Distribution Studies over the Ni3Mo1@C Catalyst

Reaction temperature has a great effect on the hydrodeoxygenation reaction and product distribution. The guaiacol conversion and product distribution over Ni3Mo1@C with an initial hydrogen pressure of 2 MPa is shown in Figure 5. It can be observed that the guaiacol conversion maintained a high proportion (around 100%) when the reaction temperature increased from 200 °C to 260 °C with a reaction time of 4 h, while the selectivity of cyclohexanol rapidly improved from 13.8 to 75.3% and the selectivity of cyclohexane exhibited a similar tendency, increasing from 0.9% to 17.6%. Interestingly, when the reaction temperatures were 200 °C and 220 °C, phenol was a main product, with relatively high selectivity levels of 63.2% and 32.1%. However, the selectivity dramatically decreased from 63.2% to 0 when the reaction temperature was enhanced to 240 °C. These results indicated that the conversion of guaiacol to cyclohexanol may be conducted at relatively high temperatures, and higher temperatures helped to facilitate the hydrotreatment of guaiacol, which was also reported in the previous studies. It was very common for high temperatures to easily contribute to the dearomatization of the benzene ring [2]. Sels et al. reported that higher reaction temperatures (e.g., 250 °C or 300 °C) favored the conversion of guaiacol to cyclohexanol [46]. Tomishige et al. also confirmed that a high selectivity of cyclohexanol could be obtained at a high reaction temperature during the HDO of guaiacol [47]. Taken together, in this work, the selectivity of cyclohexanol at 260 °C was slightly higher compared with 240 °C; however, the difference is little. Considering the energy consumption, we finally chose the 240 °C as the optimum reaction temperature over the Ni3Mo1@C catalyst.

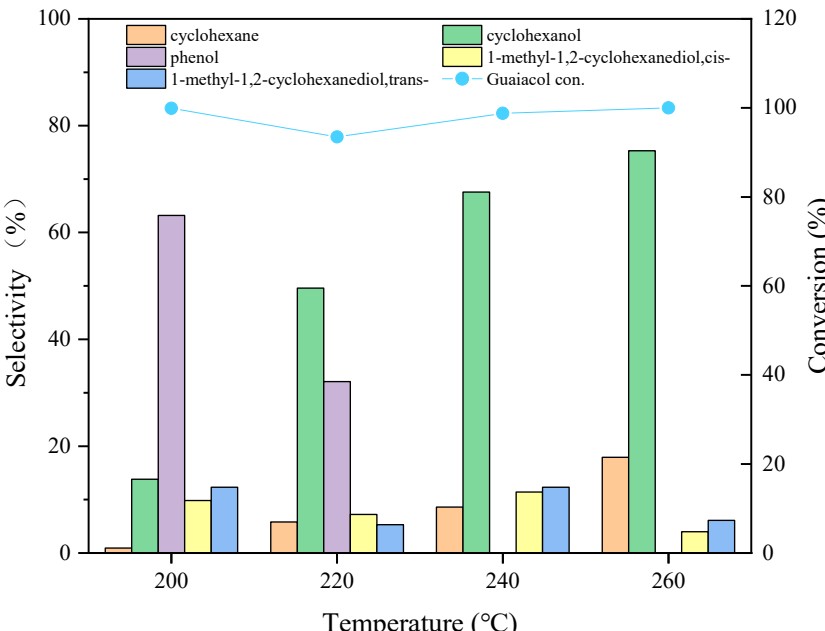

**Figure 5.** Guaiacol conversion and product distribution at different temperatures. Reaction conditions: Ni3Mo1@C catalyst, p = 2 MPa, and t = 4 h.

In this study, to study the effect of hydrogen pressure during the HDO process, the HDO of guaiacol was conducted at an initial hydrogen pressure in the range of 0 MPa to 3 MPa over the Ni3Mo1@C catalyst. As can be observed in Figure 6, the initial hydrogen pressure had little influence on the guaiacol conversion but had a significant effect on the product selectivity. The selectivity of cyclohexanol was improved from 3.8% to 74.3% with the increase of the initial hydrogen pressure from 0 MPa to 3 MPa, and the phenol selectivity dramatically decreased from 83.1% to 0 when the initial hydrogen pressure was enhanced to 2 MPa. Phenol was the product of the demethoxylation of guaiacol and cyclohexanol was the further hydrogenation product of phenol, indicating that high hydrogen pressure was beneficial to the hydrogenation of the benzene ring. In the HDO process, hydrogen was a reactant; therefore, increasing hydrogen content could favor the hydrodeoxygenation reaction due to the promoting effect of excessive hydrogen on the reaction. In addition, the solubility of hydrogen can be improved with the increase of hydrogen pressure, which makes it easier for hydrogen to arrive at the activity site of catalysts during the HDO process. These reasons are in agreement with previous study [6]. Taken together and considering that there existed little selectivity difference between 2 MPa and 3 MPa, we finally chose 2 MPa as the optimum hydrogen pressure.

In addition to reaction temperature and hydrogen pressure, the catalytic activity was also greatly influenced by reaction time. It was reported by Sels et al. that hydrodeoxygenation and repolymerization would occur simultaneously when the reaction time is less than 3 h and reaction temperature is less than 200 °C. Herein, a longer reaction time (e.g., 4 h, 6 h, and 8 h) was investigated over Ni3Mo1@C at 240 °C to study the change of product distribution. As can be seen in Figure 7, the conversion of guaiacol was maintained at almost 100%, which did not change a lot. However, there was an improvement of cyclohexane selectivity with the increase in reaction time, while the selectivity of cyclohexanol and total 1-methyl-1,2-cyclohexanediol (cis- and trans-) decreased. As was reported by Tomishige and Sels, the HDO of guaiacol to cyclohexanol is a complex process, including direct hydrogenation, hydrodeoxygenation, demethoxylation, and so on. Therefore, a proper reaction time is crucial to the HDO of guaiacol. Taking the selectivity of cyclohexanol and cyclohexane into consideration, 4 h was finally chosen as the optimum reaction time for the hydrodeoxygenation process.

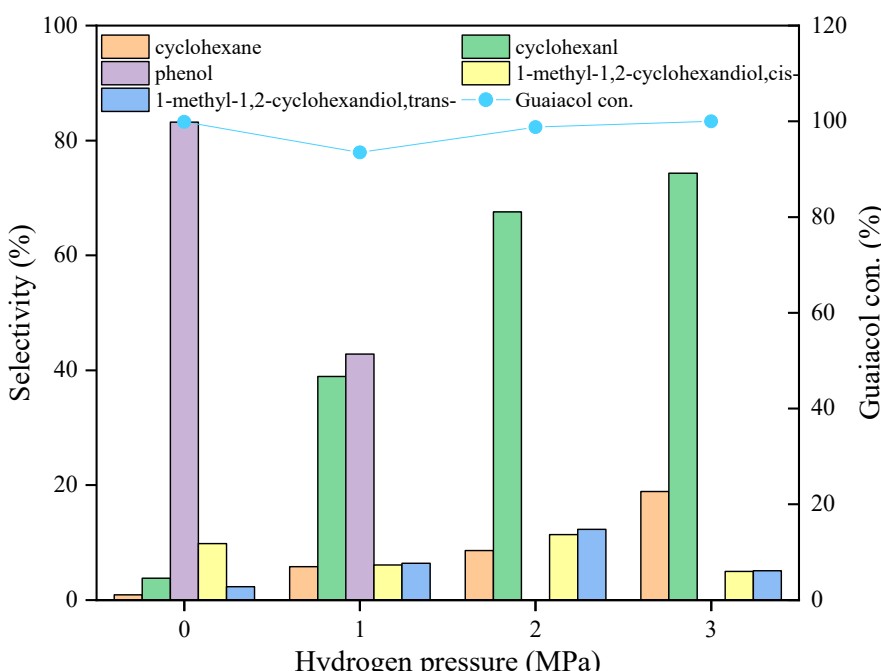

**Figure 6.** Guaiacol conversion and product distribution at different hydrogen pressures. Reaction conditions: Ni3Mo1@C catalyst, T = 240 °C, and t = 4 h.

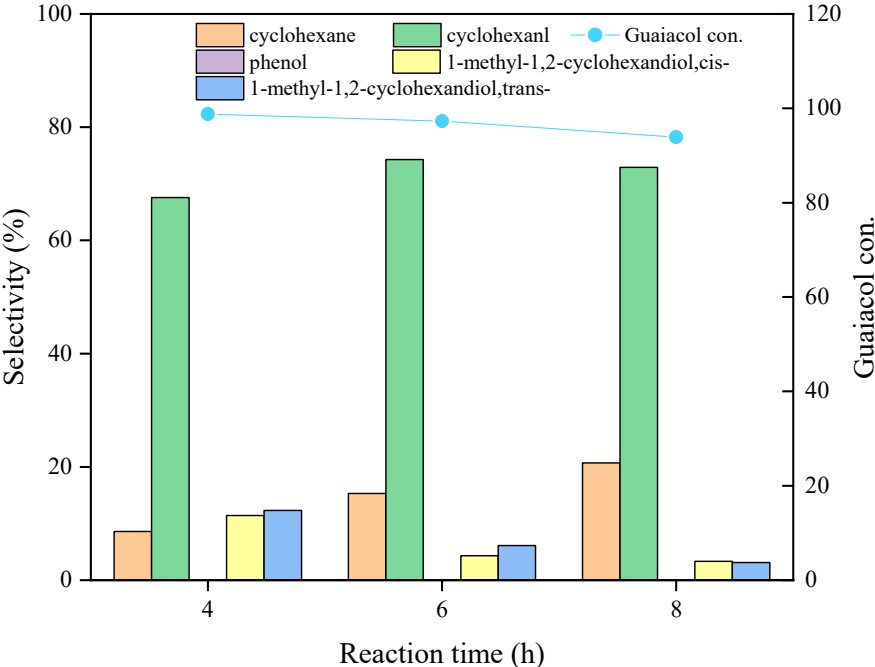

**Figure 7.** Guaiacol conversion and product distribution at different reaction times. Reaction conditions: Ni3Mo1@C catalyst, T = 240 °C, and p = 2 MPa.

To confirm the above protocol, the catalytic HDO of a series of lignin-derived monomers was studied. It was found that Ni3Mo1@C showed good performance for the HDO of different lignin-derived monomers. Phenol could be easily transformed to cyclohexanol under the optimum conditions, with a high selectivity toward cyclohexanol (Table 3, entry 2, 96%). When O-methylphenol and o-diphenol were used as substrates in the present catalytic system, 45% and 58% cyclohexanol was obtained, indicating that substitution groups on the benzene ring will reduce the efficiency of hydrodeoxygenation (Table 3,

entries 3-4). For (E)-4-(3-hydroxyprop-1-en-1-yl)phenol, it seemed difficult for its HDO to cyclohexanol in the present catalytic system. Although the conversion could be achieved at 68%, the cyclohexanol selectivity was still very low (about 16%, Table 3, entry 5), as it was difficult for the cleavage of the C-C bond in the present system. In general, we might conclude that the Ni3Mo1@C catalyst showed good catalytic HDO for most of the lignin-derived phenols, but further studies are still needed.

**Table 3.** Evaluation of the substrate scope in the HDO process over Ni3Mo1@C.

| Entry | Substrate | T(°C)/t(h)/P(MPa) | Conversion | Selectivity of Products |
|-------|-----------|-------------------|------------|-------------------------|
| 1 | OH / OMe | 240/4/2 | 100% | OH — 66% |
| 2 | OH | 240/4/2 | 100% | OH — 96% |
| 3 | OH / CH3 | 240/4/2 | 79% | OH 34% / OH 45% |
| 4 | OH / OH | 240/4/2 | 88% | OH/OH 30% / OH 58% |
| 5 | OH / (E)-hydroxypropenyl | 240/4/2 | 68% | OH 50% / OH 18% |

The reusability of the Ni3Mo1@C catalyst was also investigated. Herein, a batch of Ni3Mo1@C catalysts was used repeatedly for the HDO of guaiacol at 240 °C for 2 h with an initial H2 pressure of 2 MPa. The results, seen in Figure 8, indicated that the catalyst still showed good catalytic activity after four uses, with a subtle decrease of cyclohexanol selectivity.

Two main pathways could be found in our catalytic system, as presented in Figure 9. 1-methyl-1,2-cyclohexanediol (trans- and cis-) was detected as the intermediate during the HDO of guaiacol (pathway I), followed by the dehydration to afford 2-methyl-cyclohexanol (cis- and trans-). According to the product distribution in Figures 5–7, we deduced that the HDO of guaiacol over Ni3Mo1@C might undergo a reaction path, according to pathway II, through the formation of phenol as the main intermediate. Further, a very small amount of cyclohexane was observed in the reaction products, which was the further hydrogenation product of cyclohexanol.

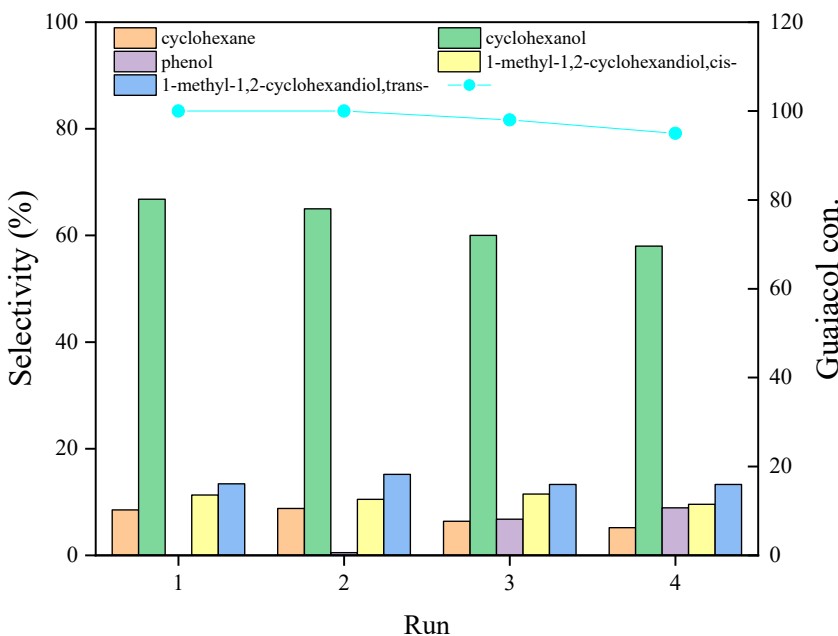

**Figure 8.** Reusability of the Ni3Mo1@C catalyst.

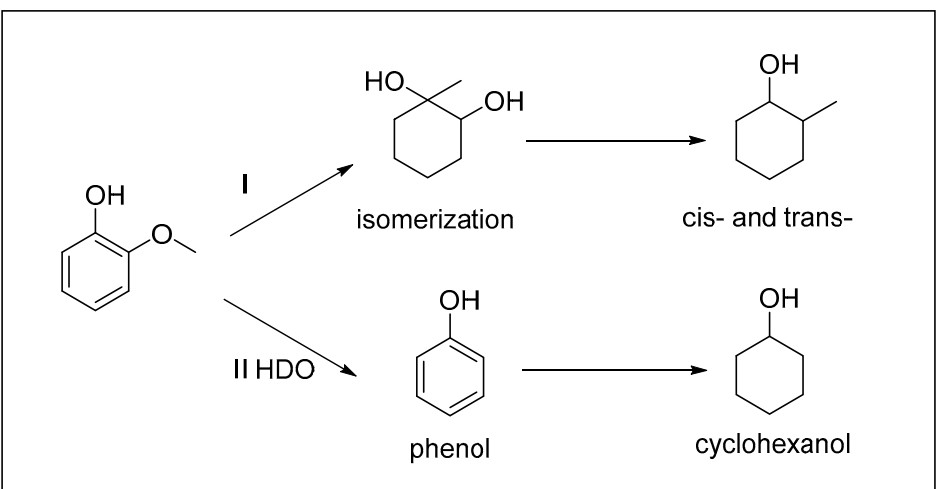

**Figure 9.** The main pathway of the HDO of guaiacol.

## 3. Experimental

### 3.1. Chemicals and Materials

In this study, all commercial chemicals were of analytic grade and directly used without any treatment. $Ni(NO_3)_2 \cdot 6H_2O$, $(NH_4)_6Mo_7O_{24} \cdot 4H_2O$, 1,3,5-benzenetricarboxylic acid, and N,N-dimethylformamide (DMF) were purchased from Aladdin Industrial Inc. (Shanghai, China). Guaiacol was purchased from Shanghai Haohong Biomedical Technology Co., Ltd. (Shanghai, China). Isopropanol was provided by China Pharmaceutical Group Co., Ltd. (Shijiazhuang, China). $N_2$ (>99.999%) and $H_2$ (>99.999%) were supplied by a local gas factory. Deionized (DI) water was used in all experiments.

### 3.2. Synthesis of Catalysts. Ni@C Catalysts

The metal–organic framework-derived, Ni-based catalysts were synthesized in two steps. An Ni-based MOF precursor was prepared in the first step by a hydrothermal process, followed by a carbonization process. In a typical run, 2.90 g of $Ni(NO_3)_2 \cdot 6H_2O$ (10 mmol) and 2.10 g of 1,3,5-benzenetricarboxlic acid (10 mmol) were dissolved in a sealed

reaction vessel containing 60 mL of DMF and stirred for 30 min. Then, the completely dissolved mixture was introduced into a 100 mL Teflon-lined autoclave and maintained at a temperature of 150 °C for 12 h. The product was filtered and then thricely washed with isopropanol to remove residuals and impurities. The obtained green solid was then dried for 12 h at 95 °C, which was donated as the Ni-MOF precursor. The obtained Ni-MOF precursor was subsequently pyrolized in a tube furnace at 500 °C for 3 h under an $N_2$ flow, and the Ni@C catalyst was obtained.

### 3.3. Ni2Mo1@C, Ni3Mo1@C, Ni4Mo1@C, and Ni5Mo1@C Catalysts

A similar procedure was employed in the preparation of the Ni2Mo1@C, Ni3Mo1@C, Ni4Mo1@C, and Ni5Mo1@C catalysts. For the preparation of Ni2Mo1@C, 1.93 g of $Ni(NO_3)_2 \cdot 6H_2O$ (6.7 mmol), 4.08 g of $(NH_4)_6Mo_7O_{24} \cdot 4H_2O$ (3.3 mmol), and 2.10 g of 1,3,5-benzenetricarboxlic acid (10 mmol) were used. For the preparation of Ni3Mo1@C, 2.18 g of $Ni(NO_3)_2 \cdot 6H_2O$ (7.5 mmol), 3.09 g of $(NH_4)_6Mo_7O_{24} \cdot 4H_2O$ (2.5 mmol), and 2.10 g of 1,3,5-benzenetricarboxlic acid (10 mmol) were used. For the preparation of Ni4Mo1@C, 2.32 g of $Ni(NO_3)_2 \cdot 6H_2O$ (8 mmol), 2.47 g of $(NH_4)_6Mo_7O_{24} \cdot 4H_2O$ (2 mmol), and 2.10 g of 1,3,5-benzenetricarboxlic acid (10 mmol) were used. For the preparation of Ni5Mo1@C, 2.42 g of $Ni(NO_3)_2 \cdot 6H_2O$ (8.3 mmol), 2.10 g of $(NH_4)_6Mo_7O_{24} \cdot 4H_2O$ (1.7 mmol), and 2.10 g of 1,3,5-benzenetricarboxlic acid (10 mmol) were used.

### 3.4. Catalyst Characterization

XRD patterns of the catalysts were performed on a Bruker D8 Advance Xray powder diffractometer using Ni-filtered Cu Kα radiation (λ = 1.5406 Å) with a scan speed of 2°/min and a scan range of 5–80° at 30 kV and 15 mA. SEM was studied using a TESCAN-VEGA3 instrument. TEM images of the catalysts were captured at 100 kV on a JEOL 1010 TEM. The content of the Ni and Co was determined by Laser Ablation Inductively Coupled Plasma Mass Spectrum (LA-ICP-MS, Applied Spectra, Fremont, CA, USA). The acidity of the catalysts was characterized by NH3-TPD with a 1.341 mmol/ min measured flow rate on a Micromeritics AutoChem 2920 instrument. Py-IR was performed on a Thermo Fisher Nicolet iS50 (Waltham, MA, USA) to investigate the quantities of acid sites on the NiMo@C catalysts.

### 3.5. Catalytical Hydrodeoxygenation of Guaiacol

All the guaiacol hydrodeoxygenation reactions were carried out in a 25 mL stainless autoclave with a magnetic stirrer. In a typical run, 100 mg of guaiacol, 10 mg of a catalyst, and 10 mL of isopropanol were added into the reactor. The reactor was purged with $H_2$ for three times to exhaust the air, and then the reaction was carried out at a desired temperature, $H_2$ pressure, and reaction time, and stirred at 600 rpm. After the reaction, the reactor was cooled in an ice bath and the reaction mixture was filtered to remove the catalyst in order to collect the liquid product. The $H_2$ pressure was changed from 0 to 3 MPa, the reaction temperatures were varied from 200 °C to 260 °C, and the reaction time was changed from 4 h to 8 h. Conversion and corresponding product yields were calculated in total molar basis, and the defined equations are listed as follows:

$$\text{conversion} = \frac{\text{mole of reacted guaiacol}}{\text{total mole of guaiacol}} \times 100\% \tag{1}$$

$$\text{yield of cyclohexanol} = \frac{\text{mole of cyclohexanol}}{\text{total mole of guaiacol}} \times 100\% \tag{2}$$

$$\text{yield of ethylcyclohexane} = \frac{\text{mole of ethylcyclohexanol}}{\text{total mole of guaiacol}} \times 100\% \tag{3}$$

Additionally, the $H/C_{eff}$ ratio represents the effective hydrogen to carbon ratio, which was defined as the molar fractions of atomic constituents in the biomass from the following equation: $H/C_{eff} = (H-_2O-_3N-2S)/C$. For ethanol ($C_2H_5OH$), its H/C value is 2.0.

## 4. Conclusions

In this work, the catalytical hydrodeoxygenation of guaiacol was carried out over metal–organic framework (MOF) -derived NiMo@C catalysts at a moderate condition. Among these catalysts, the Ni3Mo1@C catalyst exhibited a better guaiacol conversion of 97.3%, with cyclohexanol as the main product, under the optional conditions of 240 °C, 4 h, and 2 MPa $H_2$. The catalyst characterization results confirmed the synergistic effect between metal Ni and Mo. The addition of Mo decreased the particle size, improved the metal dispersion, and strengthened the acidity, which improved catalytic activity in the HDO process. Above all, bimetallic NiMo-MOF-derived catalysts exhibited better catalytic activity for the hydrodeoxygenation of guaiacol, which provides a novel strategy for bio-oil upgrading, instead of noble metal.

**Supplementary Materials:** The following are available at: https://www.mdpi.com/article/10.3390/catal12040371/s1. Figure S1: XPS of different NiMo@C catalysts.

**Author Contributions:** Formal analysis, H.X.; Funding acquisition, M.Z.; Investigation, J.L. (Junli Liu), J.J. and C.C.; Methodology, J.Z.; Resources, J.L. (Jing Li) and X.Y.; Writing—original draft, F.G. All authors have read and agreed to the published version of the manuscript.

**Funding:** This research was funded by the National Natural Science Foundation of China grant number [32071718, 31700645] and the Young scientific and technological talents promotion project of the Jiangsu Association for science and technology grant number [TJ-2021-067].

**Data Availability Statement:** Data sharing is not applicable to this article.

**Conflicts of Interest:** The authors have no conflict of interest.

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
