# Peer review of "Catalytic Hydrodeoxygenation of Guaiacol to Cyclohexanol over Bimetallic NiMo-MOF-Derived Catalysts"

_catalysts, doi:10.3390/catal12040371_

Round 1

Reviewer 1 Report

The present manuscript entitled “Catalytic hydrodeoxygenation of guaiacol to cyclohexanol over bimetallic NiMo-MOF derived catalysts” authored by Minghao Zhou et al. describes the HDO of guaiacol, a typical lignin-derived, was carried out over metal-organic framework (MOF) derived Ni-based catalysts. Furthermore, the synergistic effect between Ni and Mo was confirmed by various characterization techniques such as  ICP, XRD, SEM, TEM, and NH3-TPD analyses. In addition, the effect of reaction temperature, time, and H2 pressure during the HDO process was discussed in detail. The authors report an interesting approach and the presentation of the work is clear. The objective and justification of the work are clear, and the experimental work is significant. The study is accurate and adequate, and thus, I would recommend it for publication in Catalysts. However, certain Minor issues are detailed below to improve the quality of the manuscript.

I advise the authors to consider the following points while revising their manuscript.

Comment 1:  There are so many typographical errors such as subscripts and superscripts in the manuscript text, so authors need to correct them in the revised manuscript. English in the Manuscript should be thoroughly checked and corrected.

Comment 2:  The abstract should be edited. Please refer to some scientific achievements in the abstract and the conclusions.

Comment 3:  Some relevant references in this area are still missing in the introduction section, so include some significant relevant references from recent years.  

Comment 4:  The structure of the whole paper needs to be reopened, the results and discussion sections need to be discussed in more detail.

Comment 5: Figures 2 and 3 SEM images scale bar is not properly visible, so redraw the scale bar of SEM images manually.

Comment 6: The similarity report of the article is *36%*, while the standard of the journal is lower than *25%*. Please refer to the similarity report attached and reduce the duplication rate, so the authors need to rewrite the majority of sections (Sections 2.1 to 2.5) of the manuscript.

Author Response

Response to Reviewer 1 comments

Comment 1:  There are so many typographical errors such as subscripts and superscripts in the manuscript text, so authors need to correct them in the revised manuscript. English in the manuscript should be thoroughly checked and corrected.

Response 1: We really appreciate your comments, and apologize for our poor English. We have carefully checked the manuscript and corrected the grammatical and spelling mistakes with many typographical errors.

For example:

“3.2.1 Effect of mental composition in catalysts” was changed to “3.2.1 Effect of mentl composition in catalysts”

Comment 2:  The abstract should be edited. Please refer to some scientific achievements in the abstract and the conclusions.

Response 2: We appreciate your comments, and we improved the abstract.

Comment 3:  Some relevant references in this area are still missing in the introduction section, so include some significant relevant references from recent years.

Response 3: We understand what you mean, so we cited some relevant references about catalytic hydrodeoxygenation, which used monometallic, bimetallic or MOF-derived catalysts.

Comment 4:  The structure of the whole paper needs to be reopened, the results and discussion sections need to be discussed in more detail.

Response 4: Thank you for your comments, and we have revised the structure of the whole paper as requested by the journal. We also further discuss and analyze the results in detail.

For example, XPS and other discussion was added, which could be found below:

“The valence states of metal Ni and Mo were investigated by XPS, as shown in Figure S1. The peaks located at 869.7 and 852.4 eV corresponded to the characteristic peaks of Ni 2p1/2 and Ni 2p3/2, which were assigned to metallic Ni. The peaks located at 856.1 eV corresponded to the characteristic peaks of Ni 2p3/2, and it was assigned to nickel oxide. Small amount of nickel oxide appeared in Ni2Mo1@C catalyst, indicating that the addition of excessive Mo leads to the oxidation of metal Ni.”

Comment 5: Figures 2 and 3 SEM images scale bar is not properly visible, so redraw the scale bar of SEM images manually.

Response 5: We are very sorry for our unclear pictures, and we have already polished the figures.

Comment 6: The similarity report of the article is *36%*, while the standard of the journal is lower than *25%*. Please refer to the similarity report attached and reduce the duplication rate, so the authors need to rewrite the majority of sections (Sections 2.1 to 2.5) of the manuscript.

Response 6: The plagiarism report that the journal provides shows that the similarity of the manuscript complies with the journal's publication regulations, thus, no reduction of the manuscript's duplication is required.

Reviewer 2 Report

Could you give additional information concerning oxidation state of Ni and Mo. The authors state that Ni present as a metal. Is the same situation with Mo? XPS data could provide information about the oxidation stat of different elements.

 Are the structural and textural properties of the catalysts preserved after the reaction?

The quality of Fig 1 has to be improved.

There are some technical mistakes. For example, “3.2.1 Effect of mental composition in catalysts”

Author Response

Response to Reviewer 2 comments

Comment 1: Could you give additional information concerning oxidation state of Ni and Mo. The authors state that Ni present as a metal. Is the same situation with Mo? XPS data could provide information about the oxidation stat of different elements.

Are the structural and textural properties of the catalysts preserved after the reaction?

Response 1: XPS analysis was provided in the supporting information, and the improved discussion was below:

“The valence states of metal Ni and Mo were investigated by XPS, as shown in Figure S1. The peaks located at 869.7 and 852.4 eV corresponded to the characteristic peaks of Ni 2p1/2 and Ni 2p3/2, which were assigned to metallic Ni. The peaks located at 856.1 eV corresponded to the characteristic peaks of Ni 2p3/2, and it was assigned to nickel oxide. Small amount of nickel oxide appeared in Ni2Mo1@C catalyst, indicating that the addition of excessive Mo leads to the oxidation of metal Ni.”

Comment 2: The quality of Fig 1 has to be improved.

Response 2: We are very sorry for our low-quality figure, and we have improved the quality of Fig 1 in the paper.

Comment 3: There are some technical mistakes. For example, “3.2.1 Effect of mental composition in catalysts”

Response 3: Thanks for your kind suggestion, we have polished our manuscript and such spelling error was checked throughout the manuscript.

Reviewer 3 Report

The paper from Zhou et al. is a follow-up of the paper Chen, C. et al. 2021 (doi:10.1021/acssuschemeng.1c05273) released by the same Authors. The catalysts here reported (one single-metal and four Ni-Mo at different stoichiometric ratio) are the same therein presented and their characterization (with the exception of the TEM survey) has been already detailed in Chen, C. et al. 2021 (which also include the characterization of the pristine MOFs). The only difference with the parent paper is a focus on one monomer from lignin (guaiacol). Moreover, the performance toward guaiacol conversion has been already partially reported in Chen, C. et al.

Given this premise, I found most of the data reported in the submitted paper not original and then not suitable to be reported. Moreover, the novel data from guaiacol conversion (conversion yield at different temperature, H2 pressure and reaction times) are non sufficient for the high-quality target of Catalysts journal.

For the reasons above disclosed the paper is not acceptable for publication in the present form.

My advise for a future submission is to enrich the paper by including the results from the other lignin relevant monomers (e.g. phenol, syringol) and to stress the originality of the paper.

In the following I would like to provide comments to assist the Authors for future submissions.

Minor comments:

Since the calcination was conducted in inert atmosphere (probably here the word “pyrolized” would be preferred), it is important to have the knowledge of how much C material remains inside the MOF-based catalyst.

SEM inspections: please provide readible scale-bar also for SEM images (as it has been done for elemental mappings)

“… which accounts for 15-30wt% masses and ap-36 proximately 40% energy of lignocellulose, offering an opportunity to convert into value-37 add chemicals and liquid fuel”: please specify what exactly is intended here as “energy” (calorific value, specific heat…)

Author Response

Response to Reviewer 4 comments

The paper from Zhou et al. is a follow-up of the paper Chen, C. et al. 2021 (doi:10.1021/acssuschemeng.1c05273) released by the same Authors. The catalysts here reported (one single-metal and four Ni-Mo at different stoichiometric ratio) are the same therein presented and their characterization (with the exception of the TEM survey) has been already detailed in Chen, C. et al. 2021 (which also include the characterization of the pristine MOFs). The only difference with the parent paper is a focus on one monomer from lignin (guaiacol). Moreover, the performance toward guaiacol conversion has been already partially reported in Chen, C. et al. Given this premise, I found most of the data reported in the submitted paper not original and then not suitable to be reported. Moreover, the novel data from guaiacol conversion (conversion yield at different temperature, H2 pressure and reaction times) are non-sufficient for the high-quality target of Catalysts journal. For the reasons above disclosed the paper is not acceptable for publication in the present form.

My advice for a future submission is to enrich the paper by including the results from the other lignin relevant monomers (e.g. phenol, syringol) and to stress the originality of the paper.

We added several monomers (e.g. phenol, syringol) in our manuscript, and the discussion could be found below:

To confirm the above protocol, the catalytic HDO of a series of lignin-derived monomers were studied. It could be found that Ni3Mo1@C showed good performance for the HDO of different lignin-derived monomers. Phenol could be easily transformed to cyclohexanol under the optimum conditions, with a high selectivity toward cyclohexanol (entry 2, 96%). when O-methylphenol and o-diphenol were used as substrate in the present catalytic system, 45% and 58% cyclohexanol was obtained, indicating that substitution groups on the benzene ring will reduce the efficiency of hydrodeoxygenation (entries 3-4). For (E)-4-(3-hydroxyprop-1-en-1-yl)phenol, it seemed difficult for its HDO to cyclohexanol in the present catalytic system. Although the conversion could be achieved at 68%, the cyclohexanol selectivity was still very low (about 16%, entry 5), as it was much difficult for the cleavage of C-C bond in the present system. In general, we might conclude that Ni3Mo1@C catalyst showed good catalytic HDO for most of lignin-derived phenols, but further studies were still in need.

Table 3 Evaluation of the substrate scope in the HDO process over Ni3Mo1@C.

In the following I would like to provide comments to assist the Authors for future submissions.

Minor comments:

Comment 1: Since the calcination was conducted in inert atmosphere (probably here the word “pyrolized” would be preferred), it is important to have the knowledge of how much C material remains inside the MOF-based catalyst.

Response 1: Thanks for your kind suggestion, the word “pyrolized” was used in our manuscript.

Comment 2: SEM inspections: please provide readible scale-bar also for SEM images (as it has been done for elemental mappings)

Response 2: Thank you for your comments, we will redraw the scale bar of SEM images manually.

Comment 3: “… which accounts for 15-30wt% masses and ap-36 proximately 40% energy of lignocellulose, offering an opportunity to convert into value-37 add chemicals and liquid fuel”: please specify what exactly is intended here as “energy” (calorific value, specific heat…)

Response 3: Biomass can store energy from sunlight in the form of chemical bonds by photosynthesis, thus, the “energy” is the solar energy stored in the form of chemical bonds.

Reviewer 4 Report

This paper describes the catalytic hydrodeoxygenation of guaiacol to cyclohexanol over NiMo@C catalysts. I reject his paper for publication based on the following reasons.

  1. The main reason for rejection is that the conclusion is not supported by the facts. The details or characterizations are not enough to support the authors’ conclusion about the higher activity of Ni3Mo1@C catalyst.
  2. The paper is like a report and most of the authors’ conclusions are without any evidence.
  3. English improvement is required. The manuscript has major grammatical and spelling mistakes with many typo errors.
  4. XRD spectrum shows a peak at 45º for Ni@C. The authors claim the appearance of a new peak at 44.3º for NiMo@C catalysts. Whereas it seems a displacement of peak appearing at 45º to lower 2θ for NiMo@C catalysts because the position of the peak is slightly shifting for Ni5Mo1@C, Ni4Mo1@C, Ni3Mo1@C, and Ni2MO1@C. The authors should investigate further and provide XPS analysis.
  5. The authors should add scale in SEM images.
  6. The images of TEM are not clear. Usually, TEM is used for magnification and to get detailed images at a magnified scale in nm. Why do the authors use a very large-scale analysis by TEM? The authors should provide magnified scaled images by TEM along with elemental analysis to confirm elemental distribution at a small scale.
  7. The inset graphs in TEM images are very difficult to see properly. These are a blur. The authors should revise the graphs to make them visible easily.
  8. The inset particle size graphs in TEM images describe the particle size of whole Ni@C and NiMo@C or of Ni, Mo metals loaded? If these particle sizes are for whole Ni@C and NiMo@C ten what are the particle sizes of Ni and Mo?
  9. The description about particle size is unclear? How can the addition of Mo control the particle size of overall NiMo@C?
  10. As described, the acidity of NiMo@C increased with an increase in Mo then why did a further increase in Mo from Ni3Mo1@C to Ni2Mo@C decrease the acidity?
  11. According to Fig. 4, there was 0% phenol for Ni5Mo1@C, Ni4Mo1@C, and Ni3Mo1@C and a further increase in Mo to Ni2Mo1@C resulted in around 80% phenol production? Only the reason for increasing particle size is not sufficient to prove the claim of the authors.
  12. According to Fig. 5, more than 60% and 40% phenol production was observed for reaction temperature 200 ºC and 220 ºC, respectively. Whereas the phenol production is 0% at the higher temperature of 240 ºC and above. Why do higher temperatures improve the selectivity of cyclohexanol? Moreover, a reduction of phenol production from more than 40-60% to 0% with a mere 20 ºC increase in temperature is not acceptable without any proper justification. The authors should justify their conclusion. Moreover, it seems a different mechanism taking place at lower and higher temperature conditions. The authors should confirm the changes in the mechanism taking place and describe the mechanisms in detail.
  13. According to Fig. 6, the hydrogen pressure up to 1 MPa showed a high production of phenol which became 0% at 2MPa and above. The plausible reason should be explained with a description of the possible reaction mechanism.
  14. According to Fig. 7, the conversion of Guaiacol decreased with the passage of time. The authors should check the durability of the catalyst for a longer time on stream. Moreover, the reproducibility and reusability of the catalysts should be checked.

Author Response

Response to Reviewer 3 comments

Comment 1: The main reason for rejection is that the conclusion is not supported by the facts. The details or characterizations are not enough to support the authors’ conclusion about the higher activity of Ni3Mo1@C catalyst. The paper is like a report and most of the authors’ conclusions are without any evidence. English improvement is required. The manuscript has major grammatical and spelling mistakes with many typo errors.

Response 1: We really appreciate your comments, and apologize for poor English. We have carefully checked the manuscript and corrected the grammatical and spelling mistakes with many typographical errors.

Comment 2: XRD spectrum shows a peak at 45º for Ni@C. The authors claim the appearance of a new peak at 44.3º for NiMo@C catalysts. Whereas it seems a displacement of peak appearing at 45º to lower 2θ for NiMo@C catalysts because the position of the peak is slightly shifting for Ni5Mo1@C, Ni4Mo1@C, Ni3Mo1@C, and Ni2MO1@C. The authors should investigate further and provide XPS analysis.

Response 2: XPS analysis was provided in the supporting information, and the improved discussion was below:

“The valence states of metal Ni and Mo were investigated by XPS, as shown in Figure S1. The peaks located at 869.7 and 852.4 eV corresponded to the characteristic peaks of Ni 2p1/2 and Ni 2p3/2, which were assigned to metallic Ni. The peaks located at 856.1 eV corresponded to the characteristic peaks of Ni 2p3/2, and it was assigned to nickel oxide. Small amount of nickel oxide appeared in Ni2Mo1@C catalyst, indicating that the addition of excessive Mo leads to the oxidation of metal Ni.”

Comment 3: The authors should add scale in SEM images.

Response 3: We are very sorry for our careless, we will add scale in SEM images.

Comment 4: The images of TEM are not clear. Usually, TEM is used for magnification and to get detailed images at a magnified scale in nm. Why do the authors use a very large-scale analysis by TEM? The authors should provide magnified scaled images by TEM along with elemental analysis to confirm elemental distribution at a small scale.

Response 4: We have improved the image of TEM and the new TEM image was below:

Comment 5: The inset graphs in TEM images are very difficult to see properly. These are a blur. The authors should revise the graphs to make them visible easily.

Response 5: We have improved the TEM image, which could be seen in Response 4.

Comment 6: The inset particle size graphs in TEM images describe the particle size of whole Ni@C and NiMo@C or of Ni, Mo metals loaded? If these particle sizes are for whole Ni@C and NiMo@C ten what are the particle sizes of Ni and Mo?

Response 6: TEM images describe the particle size of Ni and Mo metals loaded. So, it was easy to find out that the particle size of Ni decreased with the addition of Mo.

Comment 7: The description about particle size is unclear? How can the addition of Mo control the particle size of overall NiMo@C?

Response 7: We improved the description about particle size, which could be found below:

“TEM results also showed that metal Ni and Mo were uniformly dispersed in catalysts and the particle size of Ni and Mo metals loaded, which was corresponded with SEM-mapping results. It was obviously observed that the main particle size and particle distribution varied with the introduction of Mo in NiMo@C catalysts. The average particle size of Ni in monometallic Ni and bimetallic NiMo catalysts were ranged in the following order: Ni@C (24.27 nm) > Ni2Mo1@C (23.39 nm) > Ni5Mo1@C (7.18 nm) > Ni3Mo1@C (6.51 nm) > Ni4Mo1@C (6.50 nm).”

Comment 8: As described, the acidity of NiMo@C increased with an increase in Mo then why did a further increase in Mo from Ni3Mo1@C to Ni2Mo1@C decrease the acidity?

Response 8: The addition Mo in NiMo@C catalysts could increase the acidity of NiMo@C catalysts. However, the addition of Mo will not continuously increase the acidity of the catalyst. Metal Mo could only provide a certain amount of acidity, and could not continuously increase the acidity of the catalyst. Therefore, When the ratio of Ni to Mo was 3:1, it showed the highest acidity. We improved the description in the manuscript.

Comment 9: According to Fig. 4, there was 0% phenol for Ni5Mo1@C, Ni4Mo1@C, and Ni3Mo1@C and a further increase in Mo to Ni2Mo1@C resulted in around 80% phenol production? Only the reason for increasing particle size is not sufficient to prove the claim of the authors.

Response 9: In addition to the increase of particle size, the acidity of NiMo@C catalysts also played an important role in the HDO reaction. Therefore, we improved the description of this section, which could be found below:

“In addition to particle, the acidity also played an important role in the guaiacol HDO process. Ni3Mo1@C exhibited a high amount acidity amount, which showed a high selectivity of cyclohexanol with low selectivity of other byproducts”

Comment 10: According to Fig. 5, more than 60% and 40% phenol production was observed for reaction temperature 200 ºC and 220 ºC, respectively. Whereas the phenol production is 0% at the higher temperature of 240 ºC and above. Why do higher temperatures improve the selectivity of cyclohexanol? Moreover, a reduction of phenol production from more than 40-60% to 0% with a mere 20 ºC increase in temperature is not acceptable without any proper justification. The authors should justify their conclusion. Moreover, it seems a different mechanism taking place at lower and higher temperature conditions. The authors should confirm the changes in the mechanism taking place and describe the mechanisms in detail.

Response 10: Under high temperature, it is easy to lead to the dearomatization of benzene ring. It was very common in many reports, and we cited some related literature in this section. The detailed mechanism was listed below:

“Two main pathways could be found in our catalytic system, As presented in Figure 8, 1-methyl-1,2-cyclohexanediol (trans- and cis-) was detected as the intermediate during the HDO of guaiacol (pathway I), followed by the dehydration to afford 2-methyl-cyclohexanol (cis- and trans-).According to the products distribution in Fig. 5-Fig. 7, we deduced that the HDO of guaiacol over Ni3Mo1@C might undergo a reaction path according to pathway II, through the formation of phenol as main intermediate. Also, very few amount of cyclohexane was observed in the reaction products, which was the further hydrogenation product of cyclohexanol.”

Comment 11: According to Fig. 6, the hydrogen pressure up to 1 MPa showed a high production of phenol which became 0% at 2MPa and above. The plausible reason should be explained with a description of the possible reaction mechanism.

Response 11: Higher H2 pressure easily contributed to the transformation of phenol to cyclohexanol, and we also cited some related literature in this section. The detailed reaction mechanism was showed in Response 10.

Comment 12: According to Fig. 7, the conversion of Guaiacol decreased with the passage of time. The authors should check the durability of the catalyst for a longer time on stream. Moreover, the reproducibility and reusability of the catalysts should be checked.

Response 12: The reusability of the optimal Ni3Mo1@C catalyst was carried out, which could be found below:

“The reusability of the Ni3Mo1@C catalyst was also investigated, herein a batch of Ni3Mo1@C catalysts were used repeatedly for the HDO of guaiacol at 240 ℃ for 2 h with an initial H2 pressure of 2 MPa. The results in Figure 8 indicated that the catalyst still showed good catalytic activity after four times use, with a subtle decrease of cyclohexanol selectivity.”

Round 2

Reviewer 3 Report

The Authors ameliorated the paper properly addressing the comments. Before acceptance they must take care of the typos. Please check for repetitions of the acronyms definition (e.g MOF).

Reviewer 4 Report

Though the authors tried to solve minor issues of the manuscript, the main reason for rejection is that the conclusion is not supported by the facts. The details or characterizations are not enough to support the authors’ conclusion about the higher activity of Ni3Mo1@C catalyst. There are major scientific flaws in the manuscript and most of the authors’ conclusions are without any evidence.

As the paper is not scientifically appropriate, I cannot approve this paper for publication.